# Zinc Protoporphyrin-9 Potentiates the Anticancer Activity of Dihydroartemisinin

**DOI:** 10.3390/antiox12020250

**Published:** 2023-01-22

**Authors:** Yu Zhang, Xu Zhang, Bing Zhou

**Affiliations:** Shenzhen Institute of Synthetic Biology, Shenzhen Institute of Advanced Technology, Chinese Academy of Sciences, Shenzhen 518055, China

**Keywords:** ZnPPIX, artemisinin, tumor, hemin, heme oxygenase

## Abstract

Besides the clinically proven superior antimalarial activity, artemisinins (ARTs) are also associated with anticancer properties, albeit at much lower potency. Iron and heme have been proposed as possible activators of ARTs against cancer cells. Here we show that zinc protoporphyrin-9 (ZnPPIX), a heme homolog and a natural metabolite for heme synthesis during iron insufficiency, greatly enhanced the anticancer activity of dihydroartemisinin (DHA) in multiple cell lines. Using melanoma B16 and breast cancer 4T1 cells, we demonstrated ZnPPIX dramatically elevated intracellular free heme levels, accompanied by heightened reactive oxidative species (ROS) production. The tumor-suppression activity of ZnPPIX and DHA is mitigated by antioxidant vitamin E or membrane oxidation protectant ferrostatin. In vivo xenograft animal models confirmed that ZnPPIX significantly potentiated the tumor-inhibition capability of DHA while posing no apparent toxicity to the mice. The proliferating index and growth of tumors after the combinatory treatment of DHA and ZnPPIX were evidently reduced. Considering the clinical safety profiles of both DHA and ZnPPIX, their action synergy offers a promising strategy to improve the application of ARTs in our fight against cancer.

## 1. Introduction

Derived from the Chinese herb *Artemisia annua*, artemisinin (ART) is a well-known drug for its potent suppressive effect on malarial parasites [1,2]. ART is an endoperoxide sesquiterpene lactone, with the internal peroxide bridge essential for its activity. A number of ART derivatives with improved solubility and efficacy have been synthesized, including dihydroartemisinin (DHA), artemether, and artesunate [3,4]. In addition to the antimalarial effect, these ARTs display a wide range of inhibitory activities against other parasites, viruses, and cancer cells. The potential usage of ARTs in cancer therapies has recently drawn much attention. To date, it has been shown that ARTs could inhibit the growth of numerous types of cancer cell lines, including melanoma, lung, prostate, gastric, breast, renal, and CNS cancer cells [5,6]. In vivo, there is also evidence, though less abundant and often less impressive, that ARTs could impair the growth of human tumors either in mice xenograft models [7,8] or even in human clinical studies [9,10].

The mechanisms underlying ARTs’ various activities are intensively explored, and structural–functional research has demonstrated that generating cytotoxic radical species via cleavage of the endoperoxide bridge is essential for the activity of ARTs [11,12]. There are data suggesting that the antimalarial and anticancer mechanisms of ARTs may not necessarily be identical. For example, the antimalarial and anticancer activities of ART derivatives are often not correlated [13]. Nevertheless, for ARTs to work, in virtually all action scenarios, it is generally accepted that the endoperoxidic bond has to be first activated (reduced). In the case of action against cancers, some biological evidence suggests that ferrous iron (Fe^2+^) is involved in the action of artemisinin [14,15,16,17,18]. Another possible activator is heme, an abundant form of iron within the cell [19,20]. Heme can react readily with artemisinin both in vitro and in vivo. After activation, multiple pathways have been proposed mediating the action of the generated reactive oxygen species (ROS), such as weakening the integrity of the cell, inducing cell cycle arrest, apoptosis, and autophagy [12], or suppressing angiogenesis by inhibiting the secretion of VEGF, VEGFR2, and KDR/flk-1 in tumors [21], or affecting signaling pathways and transcription factors associated with tumor growth, including the AMPK pathway, the Wnt/β-catenin pathway, nitric oxide signaling, CREBP, NF-κB, and MYC/MAX [22].

Given the less-than-optimal potency of ARTs against tumors, in this work, we explored how the antitumor action of DHA could be potentiated. We attempted to modulate the intracellular heme metabolism with ZnPPIX [23] to study how the cytotoxicity of artemisinin is affected. ZnPPIX is an inhibitor of heme oxygenase (HO), and could elevate intracellular heme levels by HO-dependent and HO-independent pathways [24]. We found that in both melanoma B16 and breast cancer 4T1 cell lines, ZnPPIX could significantly strengthen the anticancer activity of DHA.

## 2. Materials and Methods

### 2.1. Cell Lines and Animals

B16-F10 and 4T1 cells, kindly gifted by Dr. Yuchuan Hong (Shenzhen Institute of Advanced Technology), were maintained in RPMI-1640, supplemented with 10% FBS and penicillin-streptomycin (100 U/mL, 100 mg/mL), in an atmosphere of 5% CO_2_ at 37 °C. Female BALB/c, C57BL/6 (4 weeks) mice were purchased from Guangdong Medical Laboratory Animal Center. All animal experiments were pre-approved by the Institutional Animal Care and Use Committee (SIAT-IACUC-221102-HCS-ZY-A2209), and conducted according to the protocols of the National Regulation of China for the Care and Use of Laboratory Animals.

### 2.2. Cytotoxicity Assays

B16 or 4T1 cells were seeded at 4000 cells/well into 96-well plates and incubated for 24 h. After removing the medium, the cells were treated with new medium supplemented with specified reagents for 24 h. DHA (MACLIN, Shanghai, China) was used at concentrations of 0.5 μM, 1 μM, 2 μM, and 5 μM, and ZnPPIX (Frontier Scientific Inc., Logan, UT, USA) at concentrations of 2 μM, 5 μM, 10 μM, and 20μM; SA (MedBio, Shanghai, China) was used at 400 μM and ALA (MACLIN, Shanghai China) at 600 μM; ferrastatin-1 (MCE, Monmouth Junction, NJ, USA) was administered at 2 μM and vitamin E (Abcam, Cambridge, UK) at 500 μM. The supernatant was removed, and the cells were treated with 2 µM Calcein-AM dissolved in 100 µL of HBSS for 10 min to identify the status of the cells. Survival of the cells was recorded by fluorescence microscopy, and the number of viable cells was counted by Image J Version 1.53t [25,26].

### 2.3. Drug Synergy Analysis

Drug combination tests and statistical analyses were performed using RStudio and GraphPad Prism. To assess potential synergy of drug pairs against 4T1 or B16 cells, we built 5∗5 drug combination landscapes using Bioconductor package “synergy finder” and its Zip model, using multiple-ray design [27]. When the synergy score is less than −10, the interaction between the two drugs may be antagonistic; From −10 to 10, the interaction between the two drugs may be cumulative; Greater than 10, the interaction between two drugs may be synergistic [28].

### 2.4. Superoxide Detection and Mitochondrial Membrane Potential Assay

4T1 or B16 cells were seeded at 20,000 cells/well into 6-well plates and incubated for 24 h. After removing the medium, the cells were treated with different drugs dissolved in HBSS for 3 h, and then the probes added.

Dihydroethidium (DHE) was used as a specific superoxide probe. Superoxide can react with DHE to produce the fluorescent product 2-hydroxyethidium (and other fluorescent superoxide non-dependent oxidation products). Fluorescence of reaction products was detected by the propidium iodide (PI) channel. 10,000 events on the target gate and samples on the flow cytometer were collected, and FSC, SSC, PI channels were selected for the parameters to be analyzed.

JC-10 (Solarbio, Beijing, China) [29] is an optimized fluorescent probe for the detection of mitochondrial membrane potential [30]. At high mitochondrial membrane potentials, JC-10 aggregates in the mitochondrial matrix, forming a polymer that produces red fluorescence. The relative ratio of red to green fluorescence is often used as a measure of the proportion of mitochondrial depolarization. Green fluorescence was detected through the FITC channel; red fluorescence was detected through the PI channel. Ten thousand events on the target gate and samples on the flow cytometer were collected, and FSC, SSC, FITC, and PI channels were selected for the parameters to be analyzed.

### 2.5. Lipid Peroxidation Test

Malondialdehyde (MDA), widely used as an indicator of lipid oxidation, can react with TBA at higher temperatures and in an acidic environment to form a red MDA-TBA adduct [31,32]. 4T1 or B16 cells were seeded at 10,000 cells/well into 6-well plates and cultivated for 24 h. The drugs were dissolved in HBSS seeds, the cells were treated for 4 h, and then collected to measure the MDA using lipid oxidation assay kit (Beyotime Biotech Inc., Shanghai, China). The MDA-TBA adduct was detected at 535 nm by colorimetric method.

### 2.6. Quantification of Heme

Heme quantification was performed essentially as previously desctibed [24,33]. 4T1 or B16 cells were seeded at 20,000 cells/well into 6-well plates and cultivated for 24 h. For the 10 μM ZnPPIX group, treatment time of 1, 2, and 3 h was performed; 3 h was used for 400 μM SA and 600 μM ALA treatment. All drugs were dissolved with HBSS to a volume of 1mL, and then added to the cells. After treatment, all HBSS solution and cells were collected, and a final concentration of 1% SDS was added, sonicated for 5 min to fully lyse and dissolve the heme. 100 μL of 1 M NaCl and 100 μL of 25% (*v*/*v*) pyridine were then added, mixed upside down and centrifuged instantaneously. An amount of 200 μL of supernatant was transferred to a flat-bottomed black 96-well plate, and the absorbance of the heme-pyridine complex was recorded at OD410 [33]. Given that ZnPPIX can also form a complex with pyridine, as the background value an equal amount of ZnPPIX was added after cell lysis in the ZnPPIX treatment group. This background value was subtracted from that of the corresponding experimental group. The content of free heme in the cell samples was calculated by a standard curve calibrated with Ferroheme (MACLIN, Shanghai, China). For each experimental group, 50 μL was taken to measure the protein content by BCA (Beyotime Biotech Inc., Shanghai, China) method.

### 2.7. In Vivo Tumor Assay

4T1 and B16 cells in logarithmic growth phase were collected and diluted in saline to 1 × 10^7^ cells/mL [34], respectively. An amount of 100 mL of cell suspension was injected subcutaneously in the left axilla of mice, at least 10 in each group. When the tumors grew to 100 cm^3^ after 5 days, the mice were grouped equally and injected intraperitoneally with corn oil (10% DMSO) in the control group, DHA 50 mg/Kg, ZnPPIX 25 mg/Kg and the same dose of DHA plus ZnPPIX. DHA and ZnPPIX were dissolved in corn oil containing 10% DMSO. Tumor volume was measured every two days for 20 days by injection. The tumor volume was calculated as 0.5 × length × width^2^. Mice were weighed every three days and euthanized after 25 days.

### 2.8. Immunohistochemistry for Ki-67 and CD31

The harvested 4T1 tumors were embedded with an optimal cutting temperature compound, frozen solidified at −80 °C and then frozen sectioned at a thickness of 10 μm. Briefly, the tumors were treated with 10% methanol + 10% hydrogen peroxide for 10 min, SUMI (250 mL solution: 0.625 g gelatin, 1.25 mL Triton X-100, 250 mL TBS) was used for antibody incubation. TBS was used as the wash buffer. After incubating overnight at 4 °C with the primary antibody, ki-67 (Abcam, Cambridge, UK) or CD31 (Beyotime Biotech Inc., Shanghai, China), the tissues were hybridized with the corresponding biotinylated antibody (Beyotime Biotech Inc., Shanghai, China) for 2 h at room temperature followed by Streptavidin-Biotin Complex (Beyotime Biotech Inc., Shanghai, China) for 1 h at room temperature. The tissues were stained with DAB staining solution for 3 min at room temperature, air-dried and stained with 0.5% hematoxylin for 5 s, washed, and decolored with purified water at room temperature. The tissue was treated with a series of concentration gradients of ethanol (50%-60%-70%-80%-90%-100%, respectively) for 5 min each and dehydrated with 100% xylene delipidation for 3 h. After drying, the slices were sealed for observation and statistics analysis. Three representative xenograft tumors from each group were analyzed. Six histologically similar areas were randomly selected from each microscopic image for analysis [35].

### 2.9. Statistical Analysis

Statistical analysis was performed with one-way ANOVA and statistical results were presented as means ± SD. Asterisks indicate critical levels of significance (* *p* < 0.05, ** *p* < 0.01, *** *p* < 0.001, **** *p* < 0.0001).

## 3. Results

### 3.1. ZnPPIX Enhances the Anticancer Activity of DHA

To test whether ZnPPIX could influence the anticancer activity of DHA, we initially screened several cell lines in hand, including B16, 4T1, Hela, HepG2, U251, MDA, and PC12. Except for uncertain results obtained with HepG2, positive effects were observed in all the others. Melanoma B16 and breast cancer 4T1 were then selected for further analysis, mainly because they are of mice origin and could be more conveniently adapted to later animal studies.

The anticancer activity of the chemicals was monitored by Calcein-AM staining in cell culture studies. As shown in Figure 1A,B, DHA incubation alone suppressed B16 proliferation in a concentration-dependent manner compared to the control group, indicating its strong anticancer effect, consistent with previous reports. No obvious inhibition on B16 cell growth was detected for treatment with ZnPPIX alone, at least with the concentration we adopted (10 μM). However, when combing ZnPPIX and DHA, B16 cells exhibited a much more repressed growth than that under the DHA-only treatment, suggesting anticancer activity of DHA can be enhanced by ZnPPIX. Almost identical results were obtained using the other cancer cell line 4T1 under a similar drug treatment scheme (Figure 1C).

To quantitatively assess the synergistic effect of the drug pair against the two cell lines, a heatmap of inhibition efficiency using different concentrations of compounds has been created (Figure 1D). The map clearly indicated that adding ZnPPIX to DHA strikingly strengthened the inhibitory effect on both types of cells (Figure 1D). Moreover, the synergy landscape generated for the compound combination showed a significant synergy between ZnPPIX and DHA against both B16 (ZIP Synergy Score = 27.246) and 4T1 (ZIP Synergy Score = 20.218) cells (Figure 1E). Altogether, these results demonstrate that the anticancer action of DHA can be much boosted by ZnPPIX.

### 3.2. Modulating Heme Biosynthesis Affects the Anticancer Activity of DHA

In chemistry, both heme and ferrous iron can reduce artemisinins. Within cancer cells, it has been reported that depending on the cell type or studies, the primary activator of DHA could be heme or ferrous. In order to investigate whether heme participated in the DHA inhibition of our cancer cells, we tried to modulate heme synthesis in the two cell lines. We first used DHA and succinylacetone (SA), a potent inhibitor of heme biosynthesis, alone or together to incubate with B16 and 4T1 cells. SA addition to the cell indeed reduced heme levels. Calcein-AM staining showed that SA treatment had no noticeable impact on the cell growth. However, when SA was used together with DHA, both B16 and 4T1 cells became less repressed compared to treatment by DHA alone, indicating DHA suppression of both cell lines can be mitigated by inhibiting heme biosynthesis (Figure 2A–C). In contrast to SA, δ-aminolevulinate (ALA) [36] is a key heme biosynthesis precursor and is supposed to elevate heme levels in the cell when in excess. Consistently, while ALA alone did not obviously affect the cell survival, ALA significantly boosted DHA-induced inhibition of both cell types (Figure 2A–C).

### 3.3. ZnPPIX Drastically Elevates Intracellular Labile Heme Levels

Given that DHA inhibitory effect on the cancers depends on heme synthesis, we next investigated whether ZnPPIX affected the labile heme level and, if yes, to what extent. We used 10 uM ZnPPIX, a concentration often adopted in many other studies. One hour after ZnPPIX incubation, the labile heme level appreciably increased. By two to three hours, the heme level climbed so much that it reached 5–10 times the original (Figure 3A,C). Therefore, ZnPPIX is a potent agent in boosting intracellular free heme levels, far more so than ALA (Figure 3B,D). This elevated heme level could significantly potentiate the tumor-inhibitory activity of DHA.

### 3.4. Antioxidants Counteract the Combined Anticancer Effect of ZnPPIX and DHA

DHA, after activation, i.e., reduction of the endoperoxidic bond, generates reactive oxygen species (ROS). ROS then exerts damage to multiple components of the cell, including membrane lipid, protein, and DNA. In order to prove that the generated ROS underlies the cancer-inhibiting effect we observed here, we tested how antioxidants against ROS might influence the activity of ZnPPIX and DHA. To this end, ferroptosis inhibitor Ferrostain-1 and ROS scavenger vitamin E (VE) [35] were each tested. Ferrostain-1 acts to protect the membrane lipid from oxidative damage, while VE is a general ROS scavenger. Examination of the cell survival ratio showed that compared to the control group, Ferrostain-1 by itself had no apparent influence on the cell, while VE exhibited mild suppression on both types of cancer cells (Figure 4A,B). Nevertheless, the inhibitory effects of ZnPPIX plus DHA were greatly reduced when Ferrostain-1 or VE was additionally supplemented (Figure 4A, B), indicating that antioxidants can strongly antagonize the strong suppressive effect of ZnPPIX and DHA in cancer cells.

We further monitored ROS production and damages the cells experienced during these actions. As shown in Figure 4C, single use of ZnPPIX or DHA generated more ROS damage than the control, as reflected by the MDA assay. In contrast, combined treatment with both drugs inflicted much more lipid damage than each agent (Figure 4C), supporting a synergistic effect of the two drugs in accumulating harm inside the cancer cells. However, the ZnPPIX promotive effect was much ameliorated when heme biosynthesis inhibitor SA, antioxidants Ferrostain-1 or VE was used under ZnPPIX and DHA treatment, reflecting that these agents counteracted the actions of ZnPPIX and DHA in producing oxidative damage. Direct ROS detection by flow cytometry (FCM) using DCF probe also supported a synergistic effect of ZnPPIX and DHA in ROS production. Again the effects can be attenuated by ROS scavengers such as Ferrostain-1 and VE (Figure 4D,E). In parallel, FCM studies on membrane potential using JC-10 probe revealed that the combined application of ZnPPIX and DHA strongly depolarized the mitochondrial membrane in both cell lines, which could be weakened by either Ferrostain-1 or VE (Figure 4D,E). Collectively, these data indicate that inhibiting ROS generation by antioxidants is able to efficiently counteract the combined anticancer effect of ZnPPIX and DHA.

### 3.5. ZnPPIX Functions Synergistically with DHA against Tumors In Vivo

To confirm whether the synergistic anticancer effect of ZnPPIX and DHA exists in vivo, we lastly carried out animal studies using either drug or both to treat tumor model mice. We first established a melanoma B16 mice model. Four-week-old mice were separately injected with ZnPPIX, DHA or both, and tumor sizes were monitored every two days. The mice were sacrificed 25 days later, and tumors were harvested. As shown in Figure 5A, B16 model mice injected with single drugs developed relatively smaller tumors than the control vehicle group, suggesting each drug alone possesses certain anti-tumor abilities. Strikingly, mice injected with both ZnPPIX and DHA formed tumors much reduced in size than those with single applications of each compound, indicating a significantly enhanced anti-tumor effect when the two drugs were combined. A time course measuring the tumor growth was presented in Figure 5B, consistent with the results in Figure 5A. Similar results were independently confirmed with breast cancer 4T1 model mice (Figure 5D,E). Notably, injection of these drugs showed no obvious impact on mice weight gain compared to the control (Figure 5C,F). Further analyses revealed that, compared to tumors from controls, tumors after DHA+ZnPPIX treatment had a trend of decreasing numbers of blood vessels (Figure 5G,H), as marked by CD31, and a lower percentage of proliferating cells, as indicated by reduced Ki-67 expression (Figure 5I,J). This may partially underlie their slower growth [35]. Altogether, these data from animal studies support the combinatory use of ZnPPIX and DHA is significantly more effective in suppressing tumor development in vivo.

## 4. Discussion

ARTs have been widely utilized as an antimalarial drug for many years. However, some other biological activities were also reported, among which anticancer activity was highlighted the most. Mechanisms underlying these activities are still controversial. In terms of heme and ARTs, boosting intracellular free heme levels may augment the potency of ARTs against many types of cells but reduce their antimalarial action [24,37], consistent with results summarized in the budding yeast [37]. Though both Fe and heme have been reported underlying the activation of ARTs in their antitumor actions, our results with many cell types suggest in most cases, heme plays at least a role or even a critical role in the action of DHA, as most cell types responded more or less to the enhancing action of ZnPPIX.

The heme-elevating effect of ZnPPIX is robust. ZnPPIX differs from heme in that the center lies a Zn^2+^ instead of Fe^2+^. During heme synthesis, the last step involves Fe^2+^ incorporation; when the iron is deficient, Zn^2+^ may take its place. Thus, ZnPPIX is a natural metabolite and generally poses little harm. As an analog of heme, ZnPPIX is an inhibitor of HO, therefore, it can decrease heme breakdown and increase the level of labile heme. In malarial parasites, which lack functional HO, ZnPPIX also significantly increases labile heme levels, indicating ZnPPIX could elevate labile heme levels additionally with HO-independent ways. Since ZnPPIX adopts highly similar conformation with heme, we hypothesize it could substitute or compete with heme in protein binding, releasing an extra amount of labile heme. These two avenues combined make ZnPPIX a highly efficient labile heme booster. Because both ZnPPIX and DHA are clinically proven safe molecules, our studies offer a promising strategy to enhance the potential application of ARTs against cancers or in other uses.

Ferrostatin is often used in ferroptosis studies as it is an efficient protector against membrane oxidative damage. Our results showing that ferrostatin can significantly reduce the inhibitory action of ZnPPIX and DHA, suggest that a prime target of their combined action might be the membrane. Consistently, it has been reported that ferroptosis is involved in ARTs-mediated tumor inhibition [38,39].

While heme increase stimulates the potency of DHA against cancers, it appears not to be the case in malarial inhibition. We reported recently that elevating labile heme levels in malarial parasites, on the contrary, impedes the action of ARTs [24]. This brings us back to the central issue of action mechanism of ARTs. Our previous work in the budding yeast revealed that there are at least two kinds of intracellular action for ARTs [37]. ARTs can react with intracellular heme and produce ROS, which could potentially damage intracellular processes. This reaction requires higher amounts of ARTs, since cells are normally equipped with certain levels of antioxidative capability. The other type of action is antimitochondrial, which is more specific and necessitates less amount of the drug [37,40]. Down-regulation of heme level significantly enhanced the second type of action but weakened the first, suggesting a competitive interaction between the two pathways [40]. We hypothesized that the antimitochondrial action of ARTs reflects their antimalarial action, and the heme-mediated action mimics the anticancer action. Altogether, the biological actions of ARTs are complex and may depend on the situation, i.e., the type of cells and the biological aspect to be examined. In terms of tumor inhibition, ARTs likely act via the heme-mediated action or the non-mitochondrial type of action.

## 5. Conclusions

Our work uncovers a potent synergistic action of ZnPPX and DHA in cancer inhibition, both in vitro and in vivo. ZnPPIX efficiently boosts intracellular labile heme levels and augments DHA’s ability in ROS production and tumor inhibition (Figure 6A,B). Considering the safety profiles of both ZnPPX and DHA, the combinatory use of these two compounds may deserve further consideration in the clinical application of ARTs against cancers.

## Figures and Tables

**Figure 1 antioxidants-12-00250-f001:**
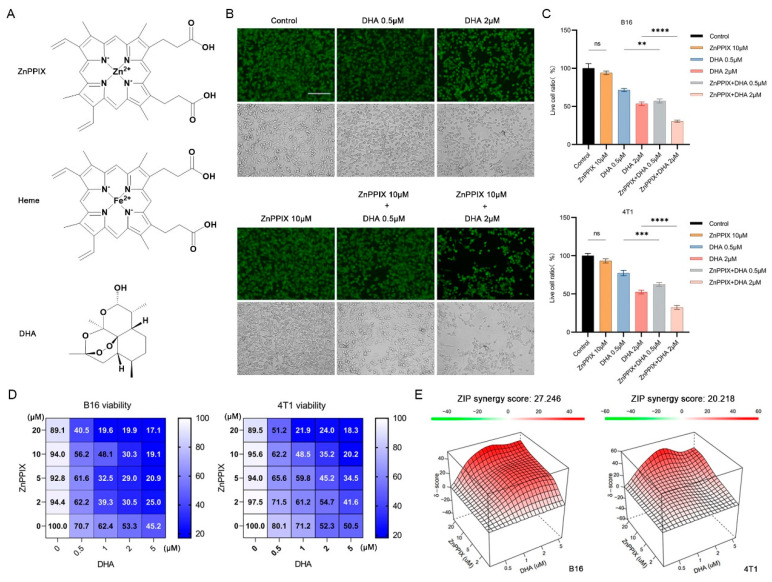
ZnPPIX enhances the anti-cancer activity of DHA. (**A**) Structures of heme, ZnPPIX, and dihydroartemisinin (DHA). (**B**) DHA inhibition on tumor cell growth is significantly potentiated in the presence of ZnPPIX. Representative fluorescence microscope images of B16 cells treated with DHA, ZnPPIX alone or in combination (bar = 125 µm). Cells were stained with Calcein-AM. Photos were taken under green fluorescence and bright field conditions, respectively. (**C**) Cytotoxicity analysis of DHA and ZnPPIX on B16 and 4T1 tumor cell lines (** *p* < 0.01, *** *p* < 0.001, **** *p* < 0.0001). (**D**) Dose–response matrix (inhibition) of B16 and 4T1 cells with DHA and ZnPPIX. (**E**) DHA and ZnPPIX inhibition synergy landscapes for B16 and 4T1 cells. Drug combination landscapes were built by “synergyfinder”. One representative replicate (with maximal synergy closest to its average value) is shown. Drug combination landscapes: *z*-axis, synergy score (ranges from −40, green, to +40, red); x/y-axes, DHA/ZnPPIX concentration range, respectively.

**Figure 2 antioxidants-12-00250-f002:**
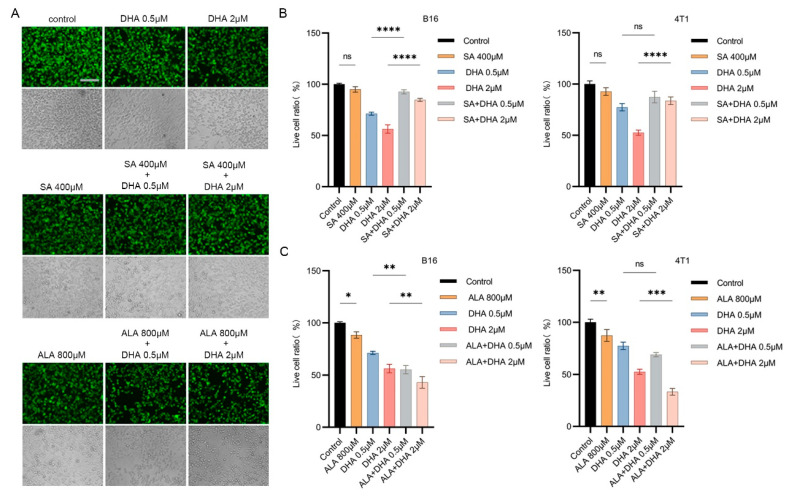
The antitumor activity of DHA depends on intracellular heme synthesis. (**A**) Representative fluorescence microscope images of B16 cells treated with DHA, SA, ALA alone, or combinations of DHA with SA or ALA (bar = 125 µm). SA (succinylacetone) is a heme inhibitor, whereas ALA is precursor of heme during its synthesis. Cells were stained with Calcein-AM. Photos were taken under green fluorescence and bright field conditions, respectively. (**B**) Cytotoxicity analysis of DHA, SA, ALA alone or DHA+SA and DHA+ALA in B16 cells. (**C**) Cytotoxicity analysis of SA, ALA, and DHA in 4T1 cells. * *p* < 0.05, ** *p* < 0.01, *** *p* < 0.001, **** *p* < 0.0001.

**Figure 3 antioxidants-12-00250-f003:**
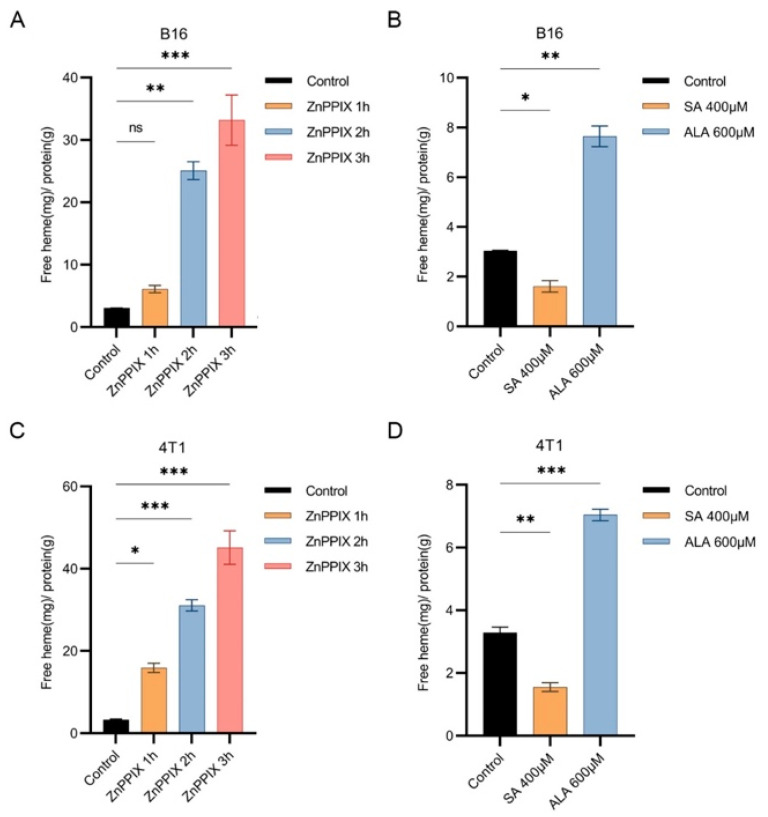
ZnPPIX elevates the labile heme concentration. (**A**) ZnPPIX increased labile heme levels in B16 cell. Two hours after treatment, a significant elevation of free heme was observed. (**B**) Altered heme levels in B16 cells after treated by SA and ALA, separately. (**C**) ZnPPIX obviously elevated labile heme levels in 4T1(C) cells 1 h after drug application. (**D**) Changed heme contents in 4T1(C) cells after SA and ALA treatments. * *p* < 0.05, ** *p* < 0.01, *** *p* < 0.001.

**Figure 4 antioxidants-12-00250-f004:**
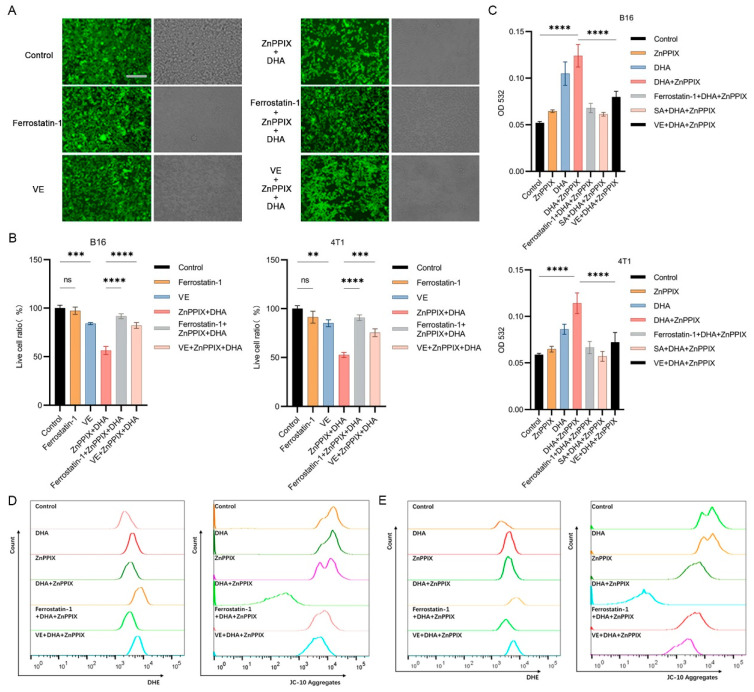
The synergistic anti-cancer effect of ZnPPIX and DHA can be suppressed by antioxidants. (**A**) Representative fluorescence microscope images of B16 cells treated with various drugs (bar = 125 µm). Ferrostatin-1 and VE both effectively neutralized ZnPPIX + DHA toxicity. Cells were stained with Calcein-AM. Photos were taken under green fluorescence and bright field conditions, respectively. (**B**) Cytotoxicity analysis of SA, ALA and ZnPPIX as measured by live cell ratios in B16 and 4T1 cells. (**C**) Detection of lipid peroxidation by MDA assay (Tecan Sunrise at OD532). (**D**,**E**) Fluorescence flow cytometry detecting ROS with DHE and JC-10 in B16 and 4T1 cells under different drug treatments. ** *p* < 0.01, *** *p* < 0.001, **** *p* < 0.0001.

**Figure 5 antioxidants-12-00250-f005:**
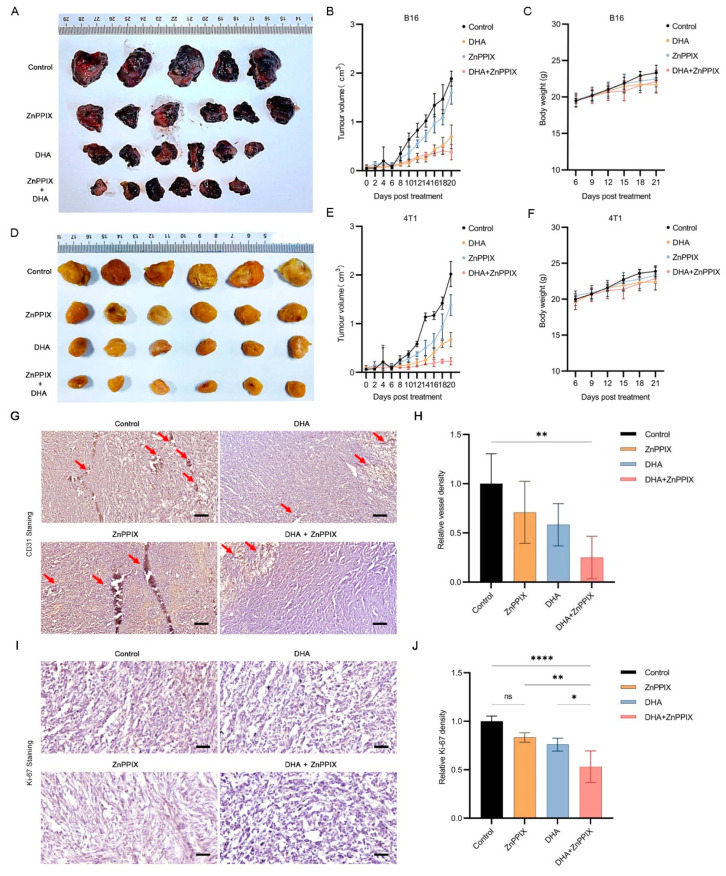
ZnPPIX and DHA act synergistically in vivo to inhibit tumorigenesis. (**A**,**D**) Photograph of tumors harvested from C57BL6 (**A**) and BABL/c (**D**) mice 25 days after B16 (**A**) or 4T1 (**D**) cells inoculation. (**B**,**E**) Tumor growth curves over 20 days after drug treatment. Tumor size was measured every 2 days. (**C**,**F**) Mice growth curves. Mice were weighed every 3 days, starting from the 6th day after inoculation. (**G**,**I**) Immunohistochemistry analysis of 4T1 tumors. Blood density analysis was performed with endothelial marker CD31 antibody staining (**G**) and proliferation with Ki-67 antibody staining (**I**). Arrows mark blood vessels. Hematoxylin staining was used for counterstain. Shown are representative pictures. Scale bars, 200 μm for CD31 staining, 50 μm for Ki-67 staining. (**I**,**J**) Relative signal intensities for data shown in (**G**,**I**), respectively. Blood vessel density (**I**) or percentages of proliferating cells (**J**) were quantified and normalized to that of controls. Data shown are means ± SD. One-way ANOVA, * *p* < 0.05, ** *p* < 0.01, **** *p* < 0.0001. In figure (**H**), when comparing specifically between DHA and DHA+ZnPPIX with student *t* test, *p* = 0.07.

**Figure 6 antioxidants-12-00250-f006:**
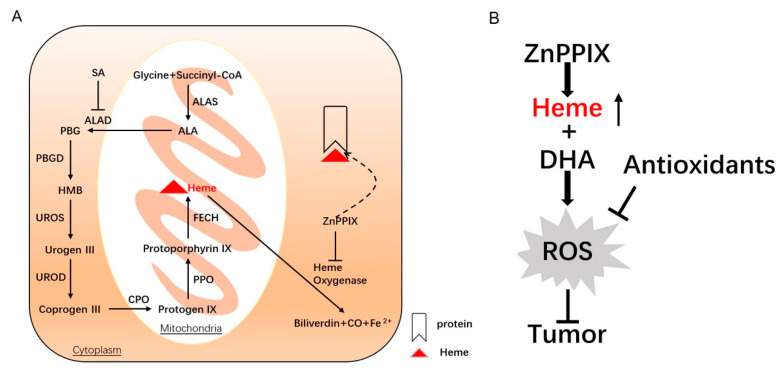
A schematic model illustrating the pathways underlying the synergistic anticancer action of ZnPPIX and DHA. (**A**) Heme synthesis and degradation. Heme synthesis starts in the mitochondrial matrix with the condensation of succinyl-CoA and glycine by ALAS to generate ALA. ALA leads to the final heme synthesis through a series of reactive processes, wherein ALAD is inhibited by SA. As an inhibitor of heme oxygenase, the heme analog ZnPPIX can reduce the degradation of heme, as well as compete with heme in protein binding, resulting in greatly heightened labile heme levels. (**B**) DHA-induced ROS generation is enhanced by ZnPPIX via heme elevation. The strengthened anti-cancer activity of DHA activity is mediated by ROS, as it can be counteracted by antioxidants that mitigate ROS accumulation.

## Data Availability

The datasets generated and analyzed during this study are included in this article. Other data are available from the corresponding author upon request.

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
