# Peer review of "Zinc Protoporphyrin-9 Potentiates the Anticancer Activity of Dihydroartemisinin"

_antioxidants, 2023, doi:10.3390/antiox12020250_

Round 1

Reviewer 1 Report

The submitted manuscript: Zinc Protoporphyrin-9 Potentiates the Anticancer Activity of Dihydroartemisinin, presents a very interesting study into the multi-benefits of artemisinins and the potentiation of their effect by Zinc Protoporphyrin-9. Yet, the following points should be addressed before further steps:

Introduction:

-In the third paragraph, it is advised to retain the objective of the study without highlighting the findings.

Materials and methods:

-Mention the source of B16-F10 and 4T1 cells.

-It is advised to reference all your methods.

-Mention make of each used material

-Make the interpretation of synergy values clearer. i.e. which values show antagonistic/ additive and synergistic relationship 

Results:

-Figure 1B and all other figures: include a side description of the two rows in green and grey.

Reviewer 2 Report

1.     First four lines of the abstract comprise introduction which is a bit long. On the other hand, the authors have not mentioned the name of techniques (methods used) in the study. Furthermore, the authors conclude that the ‘ZnPPIX significantly potentiated the tumor-inhibition capability of DHA’ however no results have been discussed. The abstract can be up to 200 words, I wonder why the authors have not used the allowed word limit to add more details of the study. Hence, I suggest the authors to reduce the introductory part of the abstract (2 lines would be sufficient), add the methodologies, discuss the most important results, and then add the conclusion.

2.     In ‘2.2. Cytotoxicity assays’ the authors state that ‘After removing the medium, the cells were treated with new medium supplemented with specified reagents for 24 h’. Please clearly mention the name of the reagents and their concentrations used.

3.     Most of the work done in this study is in vitro, although the authors did perform the ‘in vivo tumor assay’, the authors only studied the tumor size which is one aspect of the study. I suggest the authors to study the tumor biomarkers by doing ELISA/western blot, which would further cement their conclusion.

Overall, the manuscript ‘Zinc Protoporphyrin-9 Potentiates the Anticancer Activity of Dihydroartemisinin’ is a good piece of work. The results are promising, and the manuscript is worth publishing. However, there are some areas which can be improved. The abstract and most of the methodology section is too brief. I would suggest the authors to add information to make them more detailed and comprehensive. The quality of results look good and it is evident that ZnPPIX potentiates the antitumor potential of DHA. Some fluorescence images are blurry and might be replaced with clear ones. Last but not the least. The authors should study the in vivo potentiated effect of ZnPPIX and DHA at genes and molecular level to add value to the study.  

Round 2

Reviewer 2 Report

Acceptable for publication